# Making Digital Government More Inclusive: An Integrated Perspective

**Yaser Hasan Al-Mamary \*** and **Mohammad Alshallaqi**

Department of Management and Information Systems, College of Business Administration, University of Ha'il, Hail 55471, Saudi Arabia
\* Correspondence: yaser_almamary@yahoo.com

**Abstract:** This study aims to identify the key factors that can contribute to making digital government more inclusive. This study developed a research model based on integrating the theory of e-government adoption and innovation resistance theory. The empirical testing was carried out in Saudi Arabia, which is widely considered one of the most progressive nations in the Middle East in terms of digital government transformation. In total, 412 people participated in this study. This study used structural equation modeling to validate the integrated model. Based on this study's findings, this study identified the primary factors that can help make digital governments more inclusive. The most crucial elements are perceived compatibility, perceived awareness, availability of resources, perceived information quality, perceived trust, perceived functional benefits, and perceived service response. The results of this research inform government officials and policymakers in their move toward the goal of inclusive and easily accessible digital government services.

**Keywords:** structural equation modeling; digital public services; e-government adoption model; innovation resistance theory; digital inclusion

## 1. Introduction

The advent of the internet has revolutionized the operational models of public service organizations throughout the world. Citizen–government interactions evolved from basic informational websites to transactional portals and are now being reshaped by artificial intelligence and automation. Reliance on digital systems helps to bring massive efficiencies for governments. Nonetheless, such gains hinge on citizens' ability to effectively access and use digital systems, as there are many who cannot access or use such digital systems effectively.

This digital government transformation, therefore, brings with it the risk of digital exclusion, which is especially exacerbated by the reliance on 'digital by default' policies that make using digital portals mandatory (Schou and Pors 2019; Tangi et al. 2021). This risk is addressed in the literature on digital inclusion (Peeters and Widlak 2018). The literature on digital inclusion evolved from addressing questions relating to the digital divide (the 'haves' and 'have nots') (Helbig et al. 2009) to investigating the myriad ways in which people might be excluded due to socio-economic conditions (Pethig et al. 2021). The debate thus far has focused primarily on the 'user' side of the equation, investigating issues relating to physical access to technology (Okunola et al. 2017), training and capabilities building (Lee and Porumbescu 2019), and policy design (Reggi and Gil-Garcia 2021). Although addressing such issues is of paramount importance, this study takes a different approach to investigating the topic of digital inclusion. However, if access is no longer an issue, which is increasingly becoming the case, what other factors might contribute to or sustain digital exclusion?

This study adopts a novel approach to investigating this issue. It tries to pinpoint the crucial elements that might contribute to increasing digital inclusion. It draws on

innovation resistance theory (IRT) and the e-government adoption model (GAM) to flesh out the factors that might deter citizens from adopting an e-government innovation, thereby sustaining their digital exclusion. Studies on this topic in the e-government literature have yet to integrate such insightful models and develop novel insights that can advance the digital inclusion debate.

This study will utilize surveys with a variety of respondents and use a quantitative research approach to explore their perspectives on using e-government services. SEM will be used to examine the data gathered, with the frameworks of innovation resistance theory and the adoption model for e-government being applied. This study aims to improve our understanding of digital inclusion by merging innovation resistance theory and the adoption model for e-government. Finally, the conclusions drawn will guide tactics and policy proposals that support inclusive e-government designs.

## 2. Literature Review

### 2.1. From the Digital Divide to Digital Inclusion: An Overview

It has been acknowledged that the digital divide has to do with how different people, businesses, and places of residence can access resources, use computing facilities, and use information and communication tools (Norris 2001; Van Dijk 2006). It also describes a gap between individuals who possess the ability, knowledge, and skills necessary to use technologies and those who do not (DiMaggio et al. 2004; Ferro et al. 2011). The focus of policymakers' efforts to lessen these disparities has been to increase physical access to digital artifacts, such as tablets and laptops (Correa et al. 2017).

Such a focus on physical access to technology as a solution to digital divide issues has been criticized, as it overlooks fundamental issues of the ability to use technology and navigate the internet efficiently (Ebbers et al. 2016). The emphasis placed on having access to technology places an excessive amount of weight on technological fixes. Thus, there is no room for other solutions that target other equally important components of the problem, such as those pertaining to the social aspects of the digital divide (Bailey and Ngwenyama 2009; Okunola et al. 2017).

For individuals to fully reap the benefits of the internet, they must be supported in varied ways to advance from simple uses of technology, such as web browsing, to more sophisticated and value-creating forms of uses (Chipeva et al. 2018; Livingstone and Helsper 2007). Providing such forms of support requires a greater focus on issues pertaining to psychological and functional barriers (DiMaggio et al. 2004; Ram and Sheth 1989). In this spirit, the analytical purchase of the term 'digital divide' tends to fall short of encompassing such wider dimensions.

The term 'digital inclusion' evolved to shift the focus from 'divide' to the wider concept of 'inclusion.' Digital inclusion includes not only the availability of digital devices and the internet and the knowledge of how to utilize them but also tackles the socio-economic, psychological, and functional conditions that preclude the utilization of technology by disadvantaged groups (Addo and Senyo 2021). Additionally, it can be argued that the term 'digital inclusion' encompasses the process of designing digital systems in ways that ensure the inclusion of digitally disadvantaged groups (Suchowerska and McCosker 2022). This, of course, requires a nuanced understanding of how programming codes and design features, for instance, might cause digital exclusion (Park and Humphry 2019).

The concept of digital inclusion should include the degree to which various initiatives improve the ability of individuals to interact with one another and the opportunities available to them to participate in and actively engage with existing sociotechnical systems (Alam and Imran 2015). Therefore, policies for digital inclusion should make it possible for people who have been excluded from the process of digital transformation to experience new social realities that are enhanced by the presence of digital technologies.

One of the most important concerns in e-government research is ensuring access to digital government services, specifically for digitally disadvantaged populations (Hyytinen et al. 2022; Pethig et al. 2021; Reggi and Gil-Garcia 2021), especially since it has been shown

that such populations' use of digital government services falls short of expectations (Ebbers et al. 2016; Lee and Porumbescu 2019). Digitally disadvantaged populations, such as people with disabilities or older adults, are undoubtedly the most dependent on public services in modern welfare states (Alfalah 2019; Guo et al. 2022). Such digitally disadvantaged populations might be negatively affected, specifically by the advent of 'digital by default' policies that ensure the adoption of digital government services (Schou and Pors 2019). Therefore, it is imperative, especially in the digital government literature, that the topic of digital inclusion be studied and theorized more widely and from different perspectives to guide policymaking and scholarly discussions on how to advance digital government adoption. Being open to different approaches, whether they lean toward the technological (material) or the social, is important to build a more holistic picture. This study is developed in this spirit. It constructs a novel conceptual framework that integrates two prominent theories in the field—GAM and IRT—to draw out the different possible variables that make digital government more inclusive.

### 2.2. Theoretical Framework

#### 2.2.1. An Overview of the E-Government Adoption Model Theory

The GAM is a framework for identifying the factors that could affect individuals' ability to adopt e-government services across a continuum of technological sophistication (Al Mansoori et al. 2018). The GAM is the product of a marriage between the technology adoption models (TAMs), the diffusion of innovation (DOI), and the theory of planned behavior (TPB) (Choe et al. 2022; Ozkan and Kanat 2011; Zaman et al. 2021). The many drawbacks of these three approaches serve as motivations for the development of the GAM. This model has a construct that can be broken down into many variables, as shown in Figure 1.

Al Sayegh et al. (2022) stated that the setting in which e-government is adopted is the primary focus of the GAM. To better meet the requirements of a wide variety of businesses and organizations, the purpose of this model is to investigate the critical elements that play a role in the adoption of electronic government services across a spectrum of service maturity levels. For this reason, the GAM has evolved to the point that it can be used in settings as diverse as online banking and retail. Furthermore, the GAM can provide insight into the phenomenon of adopting digital government innovation from a number of unique angles, such as behavioral, technological, and cultural (Darmawan et al. 2020; Shareef et al. 2011). Despite the model's widespread adoption, pinpointing the specific factors that motivate widespread acceptance of e-government has proven difficult. This is despite the fact that the GAM model has been extensively utilized. In view of this, the GAM will be used in this study, albeit with some modifications, so that it might better serve the purposes of this investigation.

The e-government adoption model (GAM) can be criticized as being underpinned by an unescapable technological deterministic underpinning in that narrows down the complex multifaceted phenomena of e-government adoption to specific measurable variables (Grint and Woolgar 1997). This, nevertheless, does not necessarily mean that such variables are irrelevant (Sayer 1997). Also, the materiality of technology cannot be ignored (Leonardi and Bailey 2008). In the real world, the materiality of everyday life does constrain human agency in varied ways; think of how the designed features of a technology could constrain how people interact with that technological artifact (Leonardi 2011). Moreover, as Winner (1980) had shown, the designers of technology do, indeed, imbed their world views in how they build their artifacts, thereby using the materiality of technology to further their objectives. Given that technological artifacts have design features and that such features could constrain human agency, it becomes imperative to study how such material features could, in this case, impede or foster digital inclusion. In this study, we adopt what might be thought of as technologically deterministic models, not to further a technologically deterministic view, but to study and be aware of how the materiality of technology could play a role in digital inclusion. Models such as the GAM or innovation resistance theory

(IRT) can be valuable tools when adopted with care and attention to the pitfalls of technological determinism. In what follows, we explore how the GAM has been used in the e-government literature and how it might help inform the digital inclusion debate.

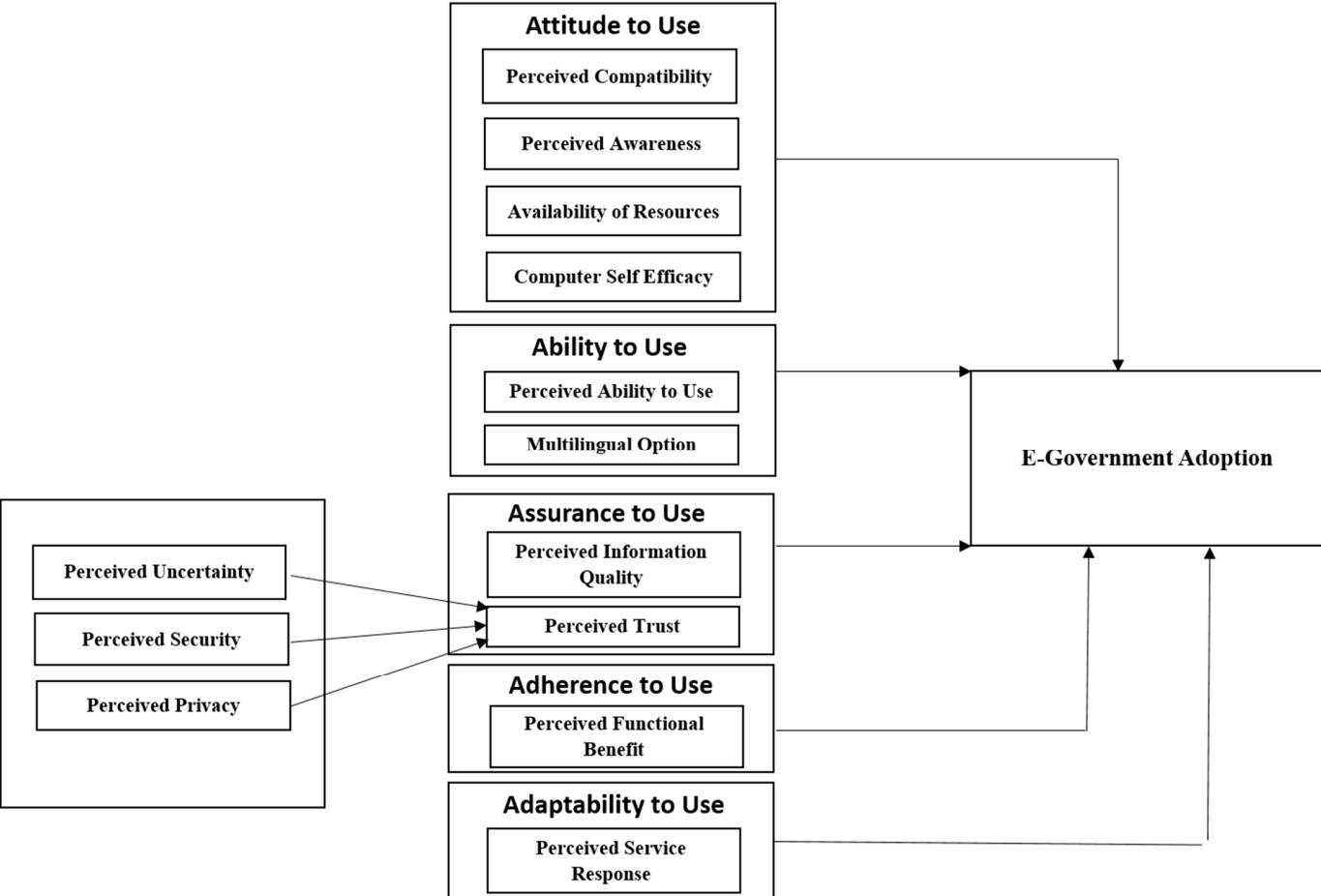

**Figure 1.** E-government adoption model. Source: Shareef et al. (2011).

2.2.2. Application of the GAM Theory to E-Government Inclusion

As a relatively new and rapidly developing topic, e-government is still in its conceptual and theoretical formative stages. Researchers from a variety of disciplines approach this issue with their own distinct hypotheses and conceptualize it in a piecemeal fashion. E-government is made up of many different aspects, some of which are social, some of which are technological, and some of which are organizational (Darmawan et al. 2020). Nevertheless, the most influential ideas concerning e-government originate not only from a socio-economic viewpoint but also from a combination of this viewpoint and a public administration viewpoint (Al Sayegh et al. 2022). Even so, they all lead to the same conclusion: setting up government systems that use information and communications technology is meant to improve the quality of service to the public as a whole.

Even if the shared goals and purposes of various countries' electronic government implementations can vary greatly, they all adhere to the same fundamental e-government value, which is that they should be citizen-focused (Lessa and Tsegaye 2019). Thus, it may be extremely important to emphasize the idea that individuals' willingness to participate in e-government is the single most important aspect in making it a reality. Although e-government is gaining popularity in several countries, it is unknown whether individuals in both developed and developing countries are willing to embrace the services that are being offered, despite evidence of tremendous growth, development, and the spread of

e-government around the world. How keen people are to use these services will impact the acceptance of e-government projects and the extent of their expansion and success.

Based on the current literature on the subject of e-government deployment, we may conclude that the GAM presented in the scholarly literature thus far is essentially conceptual. Extensive empirical investigations among actual users to validate and generalize the models are currently missing. Theorists concerned with model validation need to focus on developing their respective ontological and epistemological frameworks. Most researchers who have attempted to validate the GAM have not carried out an exhaustive analysis of the relevant body of literature and have not integrated discourses coming from a variety of perspectives, including social, organizational, and technical perspectives (Adjei-Bamfo et al. 2019). The GAM, which was discovered by Almaiah et al. (2020) and Darmawan et al. (2020) by performing an in-depth analysis of the existing research on electronic government systems, does not have a distinct theoretical grounding in terms of its methodology. When using the GAM to build an adoption strategy, the generality factor is often omitted.

Despite the widespread agreement that e-government systems can have far-reaching effects on government agencies, businesses, citizens, and the larger community, relatively little systematic and in-depth study has been conducted on the topic. Further, only a few studies have analyzed the repercussions of new technologies on government agencies, corporations, individuals, and society at large. The development of the e-government system, as well as any subsequent updates to it, adheres to certain predetermined courses, stages, and phases (Alotaibi 2020). As a result, the missions and goals of the many nations that implement e-government in their respective information and communication technology frameworks will undoubtedly be distinct. The steady development of an e-government system in each nation progresses according to a variety of varied degrees of service maturity (Khan and Krishnan 2019). Each service level is denoted by a distinct service pattern, a varying amount of technological sophistication, and a unique collection of reengineering strategies. A careful examination of these levels can reveal how the maturity of the service develops in a specific order over time.

E-government has emerged as an essential component in public sector mergers all over the world as a direct result of the improved accountability and transparency it provides (Lee-Geiller and Lee 2019; Ruvalcaba-Gomez et al. 2018). It was expected that the transition to e-government would result in significant shifts away from the conventional method of providing public services. Indeed, governments are becoming more aware of how e-government may help improve how government agencies perform and how they interact with citizens.

E-government, as defined by Siddiquee (2016), is the provision of government services via non-traditional electronic channels, such as the internet. Thus, citizens have equal access to government information and the ability to do business with the government, regardless of the time of day or location. E-government also allows for the execution of transactions at any time and from any location while adhering to equal access rules (Kumar et al. 2018). When implemented, e-government has the potential to improve ties between citizens and their governments while also bringing about transformations in the public sector. E-government is considered a novel approach to achieving the goals of local, state, national, and international economic and social development by facilitating the creation and delivery of high-quality, seamless, and integrated public services to citizens and businesses (Anshari and Hamdan 2022). This is because e-government makes it possible to deliver integrated public services seamlessly, and information and communication technology can be utilized to accomplish this goal. Anshari and Hamdan (2022) asserted that doing so can be viewed as an innovative method for the production and distribution of high-quality, hassle-free, and all-encompassing public services.

According to Bougherra et al. (2022), governments that successfully cut costs while simultaneously increasing efficiency, openness, and accountability through inclusive services have excellent and effective delivery of citizen services. Many efforts have been made by governments to incorporate e-government practices and applications into their

existing information systems and administrative procedures (Anshari and Hamdan 2022). However, because of this initiative, employees have raised their expectations regarding the capability and obligation of their government to provide new services that are effective, efficient, and modern via the internet. This has resulted in increased pressure being exerted on governments. As a result, governments in both wealthy and developing nations are increasingly adopting e-government programs (Bojang 2019) in an effort to better serve their citizens. Among these are making improvements to the government, getting closer to the people, making government agencies more efficient and effective, and cutting costs related to providing services (Apriliyanti et al. 2020; Bojang 2019).

Although some theoretical work has been performed on the adoption of e-government in developing nations, most of the emphasis has been placed on this concept in industrialized countries. This is especially true in Saudi Arabia, where significant funding has been devoted to a variety of information and communications technology endeavors. However, a number of obstacles, such as the slow pace of development and adoption of e-government programs, have limited acceptance (Alharbi et al. 2021; Nadrah et al. 2021). Thus, given that relatively little has been conducted concerning the activities of e-government in Saudi Arabia, additional research needs to be conducted.

Therefore, the current study implemented the GAM to account for factors that promote or discourage Saudi residents from using websites maintained by the Saudi government. In light of this background information, the primary objective of this research is to determine the elements that influence the attitudes of individuals in Saudi Arabia about making use of e-government services. It is expected that the findings will contribute to the formulation of plans for the delivery of electronic government services in emerging countries, such as Saudi Arabia.

### 2.2.3. Innovation Resistance Theory

As shown in Figure 2, the innovation resistance theory classifies barriers into two broad categories: functional barriers and psychological barriers (Laukkanen et al. 2009; Talwar et al. 2020). Functional barriers occur when consumers are forced to make major adjustments as a result of adopting a new innovation, whereas psychological barriers occur when the innovation's values and expectations are challenged by the new offering. Functional barriers include, but are not limited to, the utilization barrier, the value barrier, and the risk barrier. Compatibility issues with the user's existing routine, practices, or habits are the true source of user resistance. The inability to easily implement the innovation is the primary innovation in consumers' reluctance to adopt new technologies.

The value barrier is the second kind of functional barrier that might stand in the way of an innovation (Joachim et al. 2018), and it is formed by comparing the new product to the alternatives. Users will not adopt an innovation that does not provide good value in terms of the performance it offers in comparison to the price it charges. The last of the functional barriers—the risk barrier—encourages users to steer clear of unfamiliar territory whenever possible. According to Farrell and Saloner (1986), users may hold off on adopting a new technology until more information is readily available. Physical risk, financial risk, operational risk, and social risk are the four types of risks that can arise from introducing a new concept or product to the market.

All that is meant by the term 'physical risk' is that there is a chance that the innovation could cause actual physical harm to people or property (Allhoff and Henschke 2018). Economic risk is the potential for monetary loss due to the introduction of a novel innovation (Tohãnean et al. 2020). Many people and companies are waiting for the next generation of products to improve before buying them to lessen their exposure to economic risk. As a result, the risk associated with the practical application of an innovation is referred to as function risk. Consequently, the risk connected to the functionality of an innovation is referred to as function risk, whereas the risk related to the social impact of an innovation is known as social risk. One psychological barrier arises when an innovation causes consumers to reevaluate long-held cultural standards. Because of this, the amount

that a new idea requires customers to break with tradition goes up in direct proportion to how new the idea itself is.

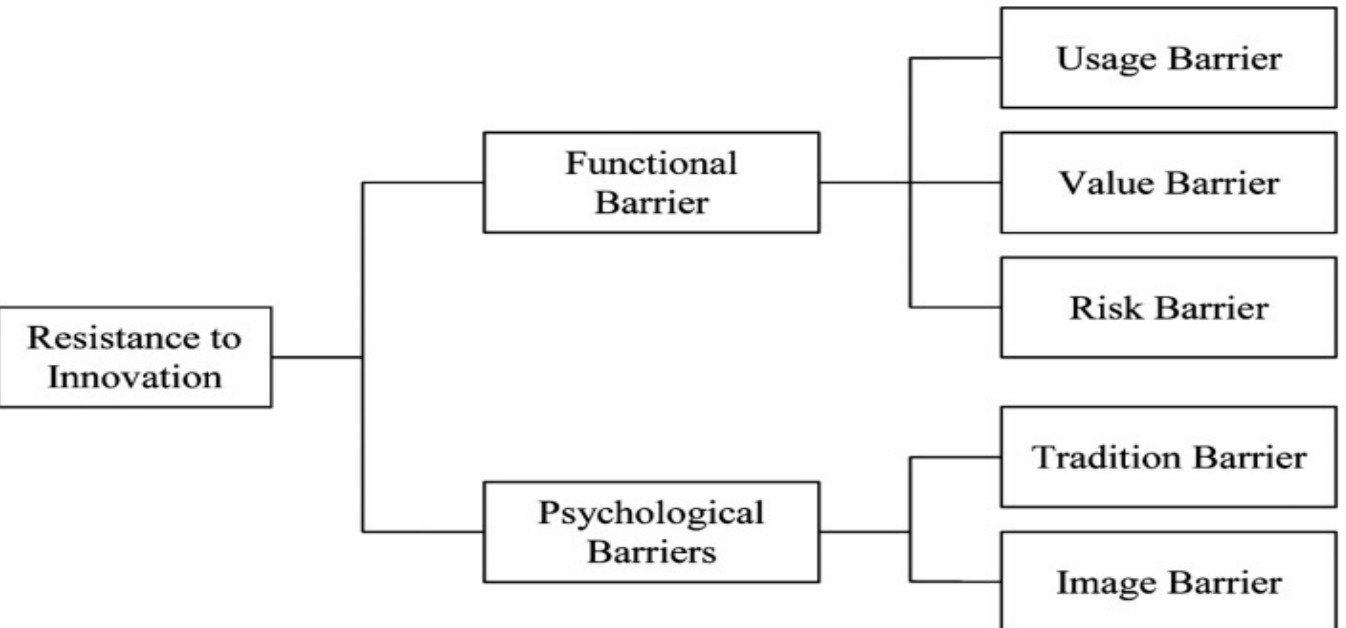

**Figure 2.** Innovation resistance theory (IRT). Source: Ram and Sheth (1989) and Luo et al. (2012).

### 2.2.4. IRT in the Field of E-Government Inclusion

The user resistance theory serves as a theoretical foundation for IRT. Its goal is to assist users in gaining a better understanding of the behaviors associated with user resistance (Kaur et al. 2020). When it comes to embracing and implementing new innovations, innovation resistance theory can be defined as behaviors that result from logical thought and deliberation in response to the potential dangers posed by changing the status quo or adopting new beliefs (Sivathanu 2018). It is possible that the level of resistance shown by users to a newly introduced feature can be an important factor in determining whether or not an e-government or e-governance initiative is successful.

When confronted with a new innovation, users may exhibit resistance-related behavior if they believe they will have to make significant changes to their way of life. As a result, there are two types of employee resistance to digital inclusion: active and passive (Appelbaum et al. 2015). The characteristics of innovations generate a type of resistance known as 'active resistance' (Castro et al. 2019), which can be investigated using the functional barriers provided by IRT. Disagreements caused by behavioral conflicts caused by the innovation's usage, value, and potential risks are the primary barriers preventing people from adopting and using new technologies (Chen et al. 2022). When long-held beliefs are called into question, innovation resistance theory posits that this elicits a passive resistance response.

IRT identifies a number of psychological barriers that can be used to study this phenomenon (Joachim et al. 2018). This theory provides a comprehensive framework for investigating employee resistance to new developments in the workplace. Consequently, current theoretical frameworks, such as models for the diffusion of innovation and the adoption of technology, do not investigate consumer resistance to newly introduced features. According to Chung and Liang (2020), researchers now have a theoretical basis for explaining why there is resistance to new innovations. IRT focuses on explaining employee responses to any digital inclusion in terms of barriers, such as long-term usage, risk involved, value addition, and overall organizational image. As more and more innovations enter the realm of e-government, this is becoming an increasingly important topic.

Implementing inclusive digital e-government is a process that spreads and is disseminated via social networks (Yang et al. 2019). This has the potential to have a variety of effects on a government's ability to compete and continue to provide excellent services while increasing its authority. According to Ashaye and Irani (2019), organizations are more likely to abandon digital government initiatives with fewer benefits if those initiatives are perceived to be detrimental to the organization's competitiveness, brand image, or finances. Despite the fact that these initiatives provide fewer benefits, they are still popular. As a result, those in charge of government innovation must investigate the primary reasons e-government projects have failed.

Identifying the factors that drive the adoption of digital public services can help the government successfully integrate digital public services for all residents. Therefore, IRT also addresses this problem by focusing on how this may be achieved by identifying the factors that influence people's use of digital public services. Further, delivering the promise of digital government through expanding digital inclusion while placing emphasis on the foundational elements of digital government may inspire more people to get involved.

Resistance to innovation may emerge after the initial stage of learning about innovative ideas or information. This is followed by the acceptance of new users on paper while attempting to avoid the testing phase. If users are unwilling or unable to participate in the evaluation step of using new items, which requires the symbolic acceptance of new goods, then a third stage of procrastination may occur. This type of procrastination is known as 'third-stage procrastination'. When new things are used, this type of procrastination occurs. Min et al. (2019) distinguished between two types of anti-innovation sentiment: unequivocal rejection and systematic procrastination. People who are resistant to change are frequently well-informed individuals whose ideas are difficult to change, regardless of the evidence presented. However, users who are hesitant to adopt new technologies may be persuaded. As a result, people will demand additional clarification before accepting them. The strategy received overwhelming support from the majority of respondents. Therefore, it is essential for academics and practitioners to have a solid understanding of innovation resistance to successfully overcome it, whether by modifying it or implementing new solutions.

Clearly, significant conceptual refinement and empirical testing will be required before IRT and the indicators that accompany it can be considered definitive. This is because IRT is in constant flux, which is the primary cause of this phenomenon. This study develops a theoretical framework for studying the e-government digital inclusion framework in Saudi Arabia, and it proposes some hypotheses based on patterns of IRT. This innovation is part of this study. The research on IRT is combined with the digital inclusion strategy implemented in Saudi Arabia's digital government transformation.

Mutiarin et al. (2019) and Ababneh and Alrefaie (2022) claimed that there has been worldwide interest in reforming the public sector through the use of information and communication technology as a medium for engaging with and serving citizens and businesses. According to the authors, this fascination has only grown since the turn of the new millennium. Ramzy and Ibrahim (2022) indicated that the term 'e-government' is used universally to describe this rapidly growing phenomenon. Over the past decade, as governments have looked for new ways to rein in spending and boost efficiency, e-government has emerged as a major innovation in public sector administration (Ababneh and Alrefaie 2022; Al-Rahmi et al. 2022; Ramzy and Ibrahim 2022). This is because government agencies are constantly seeking methods to improve internal productivity. The current era has been described using a number of different terms, such as the 'information economy', 'knowledge economy', 'digital revolution', 'new economy', 'information age', 'network society', 'digital-era governance', 'digital economy', and 'new public governance'.

A competing theory holds that the economy shifted from a managed to an entrepreneurial economy during this period. Many scholars argue that the power of enforcing laws and implementing policies is shifting from national government to subnational entities and civil society. Most Western countries have reportedly undergone this sea change in their forms of government, organizational structures, and workplace norms (Knox and Janenova 2019). As a result of this shift, the lines between public, private, and non-profit organizations are blurring, and the government is becoming more open to networked decision-making and partnership governance. To facilitate the mobilization of actors and resources beyond the state's formal setting, for example, in the implementation of public policy, new governance structures emerge. As a result, the government will have to take on new responsibilities. This shift, dubbed 'government to governance' in international research, is widely recognized (Hassan and Lee 2019). However, many argue that government is still necessary for governance to function, and that governance has not replaced government.

Multiple shifts have occurred in the past few years in the interaction between government agencies and the public, as well as within departments and teams of citizens (Abdulkareem and Ramli 2021). Due to these modifications, public administration is now performing a different set of duties. Incorporating public–private partnerships, government contracting, and project management into government administration are all examples of evidence for new public management. Entrepreneurship, of the modern kind, exists within the context of today's information society.

Among the governments in the Middle East, the Saudi government stands out as having a particularly ambitious policy goal in the realm of e-government. According to Alzahrani (2022), Lytras and Şerban (2020), and Ghazaleh and Ahmad (2018), the government of Saudi Arabia has the goal of transforming the country into a world-class information society that is open to the public around the clock seven days a week. In addition, they want to see an increase in the Kingdom's overall productivity. The Saudi government has emphasized that the needs of citizens should be at the forefront of the development of e-government (Alzahrani 2022). A national policy proposal entitled 'Proposed Framework for Quality Assessment of E-Government Portals in Saudi Arabia' was put forth in 2019 (Almurayziq and Salama 2019). This translates the concept of e-government into a facet of public administration's development. The policy ensures that the development will result in the necessary organizational changes and employee training that are required within the public administration. This policy is guided by the slogan 'simple, efficient, and secure e-government.' Almurayziq and Salama (2019) and Al-Hanawi et al. (2020) claimed that there is currently a rather high rate of internet use within the Saudi Arabian government, which may allow for and, therefore, demand, the improvement and development of governmental services on the internet. This level of internet use could also make it easier for government services that can be used on mobile devices to grow and improve.

Despite the extensive body of prior research on e-government in Saudi Arabia, there has been surprisingly little systematic research on the barriers that prevent employees from using these services. A case study of the Saudi Arabian government's implementation of e-government can provide evidence of the rapid development of e-government, despite severe financial constraints. A case study may demonstrate the importance of having appropriate managerial and technical backgrounds, planning carefully, and having competent and strict management of implementation plans based on precise goals.

Lapuente and Van de Walle (2020) indicate that when e-government programs are implemented, public sector employees may fight the new structure of their organizations. The scope of this inquiry needs to be expanded. Management and organizational problems, such as employees' defensive responses to technological change, are shown in the case of Al-Oteawi (2002) in Saudi Arabia. According to Basri (2020), who explored the state of

e-government in Saudi Arabia, reforms in this area have led to a reorganization of actor roles as a result of new practices in which the development and application of information and communication technology are becoming an increasingly central focus of business organizations, efficient citizen relationships, and the provision of public services. Reforms to the Saudi government's use of technology are proof of this. Another study found that the rhetorical prefix 'e' had become widespread within the organization but had not yet permeated the larger community at large (Nadrah et al. 2021; Al-Sakran and Alsudairi 2021; Aljarallah and Lock 2020). Hence, some parts of the idea of an electronic government have been put into practice.

Findings from a case study that investigated the introduction of a government in Saudi Arabia, with a particular emphasis placed on the interpersonal aspects of public administrators, showed that some of those working in the municipality's back office held negative views about the public system (Ali et al. 2021). The study showed that the new information and communication technology tool further blurs the line between public and private, and the authors posited this as additional proof that updating public sector organizations' information and communication technology systems is crucial (Ali et al. 2021).

The current study found that the vast bulk of earlier research on the subject of e-government conducted in the setting of Saudi Arabia focused on applying only a small number of the GAM or IRT variables in their analyses (e.g., Alfalah 2019; Alghamdi and Beloff 2014; Alonazi et al. 2018; Almukhlifi et al. 2019; Alsaif 2014); not a single study verified the complete model of the GAM or IRT. This investigation is important because it integrates two major theories of e-government to examine all the aspects that might make e-government accessible to all members of society.

Therefore, the goal of this study was to identify factors that can help make digital public services accessible to everyone. The findings are expected to offer future researchers the necessary starting point for further investigation into the topic.

### 2.3. Conceptual Model

The proposed conceptual framework was created based on the research aims, incorporating concepts from IRT and the GAM. Since the image dimension is present in both theories, we decided to use it only once in this work. This study is considered the first study in the literature on digital inclusion in e-government that combines the two theories. Structural equation modeling (SEM) tests of the integrated model were conducted with the goal of identifying the most important factors that can improve the breadth and depth of digital public services in Saudi Arabia. The conceptual framework developed to answer the study's research questions is displayed graphically in Figure 3.

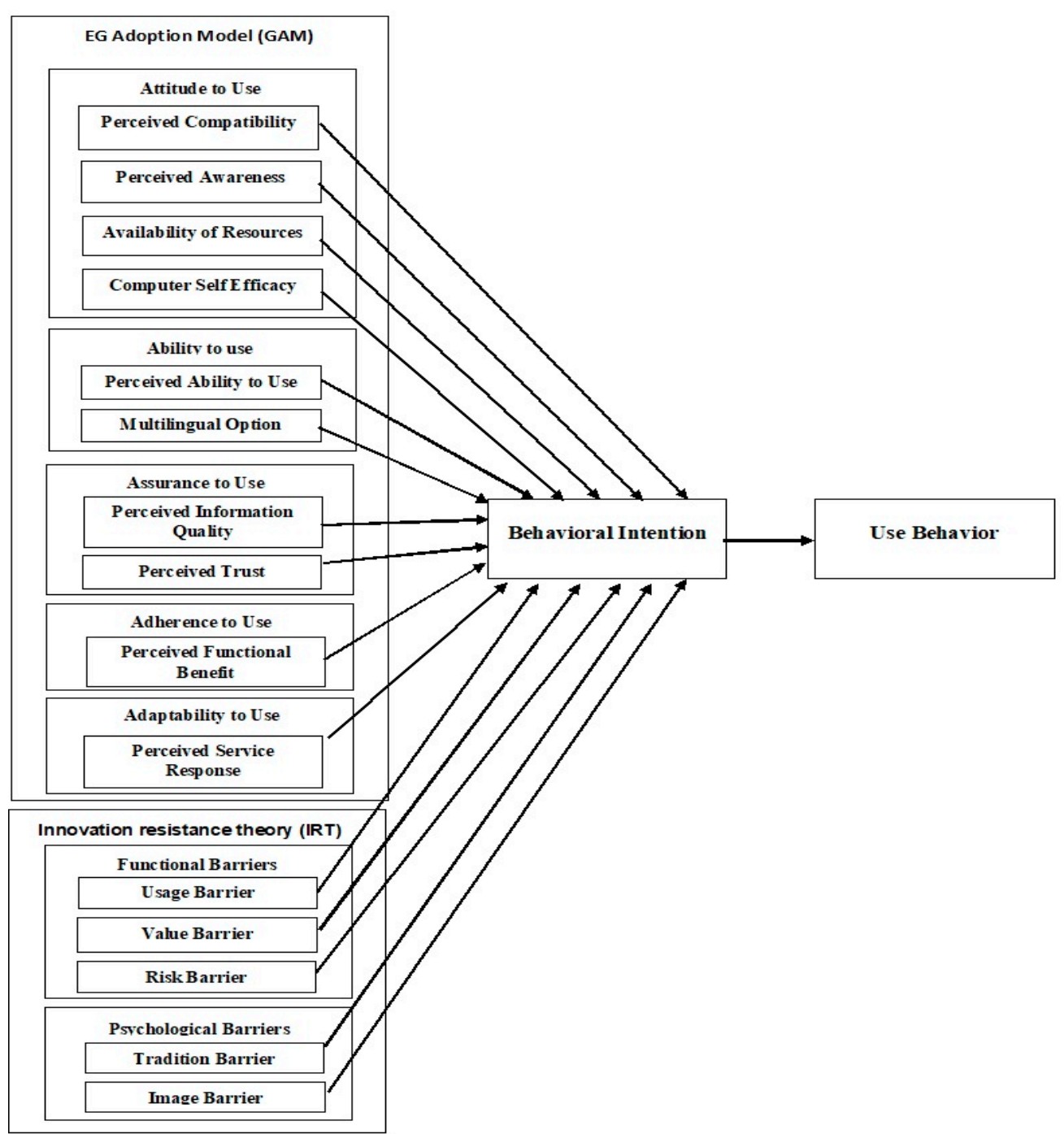

**Figure 3.** The GAM-IRT model.

### 3. Methodology

*3.1. Research Design*

The purpose of this survey study was to examine the factors that could make e-government more inclusive. By performing a thorough analysis of the previously published literature, this research obtained the fundamental understanding necessary to formulate a question that accurately expresses the study's overarching purpose: What factors, if any, make digital public services more accessible to a greater number of people? The GAM and IRT models are both prevalent in the subject of e-government and resistance to innovation, and they were both integrated in this study in some capacity.

A quantitative investigation was carried out to provide responses to the questions posed in the research. The data measuring the study variables were gathered by employing

a survey methodology. An in-depth statistical examination of the information gathered was carried out so that the study hypotheses could be confirmed. The survey items were modeled after validated instruments that had been utilized in earlier studies. A questionnaire was used as a means of collecting primary data, which was then evaluated using AMOS V21 software and the SPSS V22 statistical program after the fact. SEM was applied to this particular model to validate the model's assumptions and determine the model's fitness index. In total, 412 people participated in the study.

### 3.2. Research Instrument and Measures

In the current investigation, 72 items comprised the independent factors, the mediating variables, and the dependent variables. These items were derived from previous studies. Shareef et al. (2011) concluded that the best way to evaluate the GAM is by measuring its parameters. Furthermore, the approach used to measure IRT dimensions was adapted from Kaur et al. (2020) and Laukkanen (2016). The items for assessing behavioral intention were adapted from Kaur et al. (2020), and Camilleri's (2019) items were used to determine user behavior. On the scale for all items, 1 denotes 'strongly disagree' and 5 denotes 'strongly agree'. Before the questionnaire was made available to a wider audience, it was subjected to preliminary testing. Following the pre-test, the items were rephrased where necessary to make them suitable for use in the current study. In short, the questionnaire for the current study was adopted from previous studies, and the wording was modified to suit the current study. The questionnaire was presented in its final form to a group of experts in the field to make observations, and based on their observations, the questionnaire was formulated in its final form and distributed to the respondents.

### 3.3. Data Collection and Analysis Procedures

A Google form was used to design the survey, and responses were collected via random distribution of the survey's URL across multiple social media channels. The questionnaire link was sent through popular social media platforms used in Saudi Arabia, including WhatsApp, as most citizens use WhatsApp. Oddly, Saudi Arabia has a mobile phone penetration rate of 140% (Baabdullah et al. 2019). While most people have access to smart phones and can use social media applications, such as WhatsApp, this does not necessarily indicate that they can navigate complex e-government portals to access and procure the public services they need. By virtue of being ubiquitous, easy to use, and in many ways an essential element of daily life, mobile phones are an ideal means of collecting data. Cronbach's alpha and descriptive analysis were employed after the completed surveys were loaded into the SPSS program for coding. Cronbach's alpha was also used to determine how reliable and consistent the data were as a whole. A measurement model was initially developed for this study, which was used to test the hypotheses about the relationships.

## 4. Results
### 4.1. Demographic Profiles

The analysis in this study used responses from 412 participants. Table 1 provides a summary of the background information provided by the respondents. There was a total of 246 male participants and 166 female participants in this study, with male participants accounting for 59.7% of the sample. Overall, 79 respondents (19.2%) completed high school education, 8 respondents (1.9%) completed diploma education, 286 respondents (69.4%) completed undergraduate education, and 39 respondents (9.5%) completed postgraduate education; thus, the majority of people who participated in the study held a bachelor's degree. There were 306 respondents who were less than 30 years old (74.3%), 42 respondents between the ages of 31 and 40 (10.2%), 49 respondents between the ages of 41 and 50 (11.9%), and 15 respondents older than 50 (3.6%). Statistically, this indicates that those younger than 30 made up the bulk of the study's subjects. Table 1 also shows that the vast majority of students had at least a moderate level of computer competence (63.8%), 29.1% had a high

level of computer proficiency, and only a small minority of students (7%) had a poor level of computer proficiency.

**Table 1.** Demographic profiles.

|  |  | Frequency | Percent |
|---|---|---|---|
| Gender | Male | 246 | 59.7% |
|  | Female | 166 | 40.3% |
| Education | Higher School | 79 | 19.2% |
|  | Diploma | 8 | 1.9% |
|  | Undergraduate | 286 | 69.4% |
|  | Postgraduate | 39 | 9.5% |
| Age | Less than 30 | 306 | 74.3% |
|  | 31–40 | 42 | 10.2% |
|  | 41–50 | 49 | 11.9% |
|  | Above 50 | 15 | 3.6% |
| Level of proficiency in the use of computers | High | 120 | 29.1% |
|  | Medium | 263 | 63.8% |
|  | Low | 29 | 7.0% |

### 4.2. Measurement Model

Using the 'measurement model' part of the model, we examined the relationship between latent variables and their observable manifestations. According to Awang (2014), SEM is a confirmatory method that provides a comprehensive strategy for verifying the measurement model of latent components. The measurement models for each component were tested in the current study using AMOS version 21. However, the first measurement model's fitness indices were beyond the range proposed by Awang (2014), Al-Mamary and Alshallaqi (2022), Al-Mamary (2022c), Al-Mamary and Alraja (2022), and Rehman et al. (2022, 2023). To reach this goal, the measurement model was cleaned of any data points that did not adequately load into the relevant factors. Only the most highly loaded criteria were taken into account. This strategy helped increase the likelihood of a successful model fit. Othman et al. (2014), Al-Mamary et al. (2020), and Al-Mamary (2022b) stated that the first step is to eliminate any items with a factor loading of 0.6 or lower from the measurement model. This made the model compatible with the suggestions in Figure 4.

### 4.3. Validity and Reliability

According to Awang (2014), validity is defined as the degree to which an instrument measures the concept being targeted. Each type of validity—convergent validity, construct validity, and discriminant validity—is necessary for a measurement model to be considered valid. Tables 2 and 3 provide evidence of the convergent validity and discriminant validity of this study, respectively. The term 'reliability' is used to describe how well the measurement model can be depended upon to measure the targeted latent construct. Table 2 also shows the proof of Cronbach's alpha and CR. This is consistent with the recommendations of previous researchers (e.g., Al-Mamary et al. 2015; Al-Mamary et al. 2019; Al-Mamary 2022a; Awang 2014). Validity and reliability were achieved in the current study.

To illustrate how the extracted average variance's square roots are dispersed throughout the diagonals (Table 3), the numbers in bold show that every AVE is superior to every other connection (Al-Ghurbani et al. 2022; Al-Mamary 2021). As a result, we can claim that the constructs' discriminant validity was established.

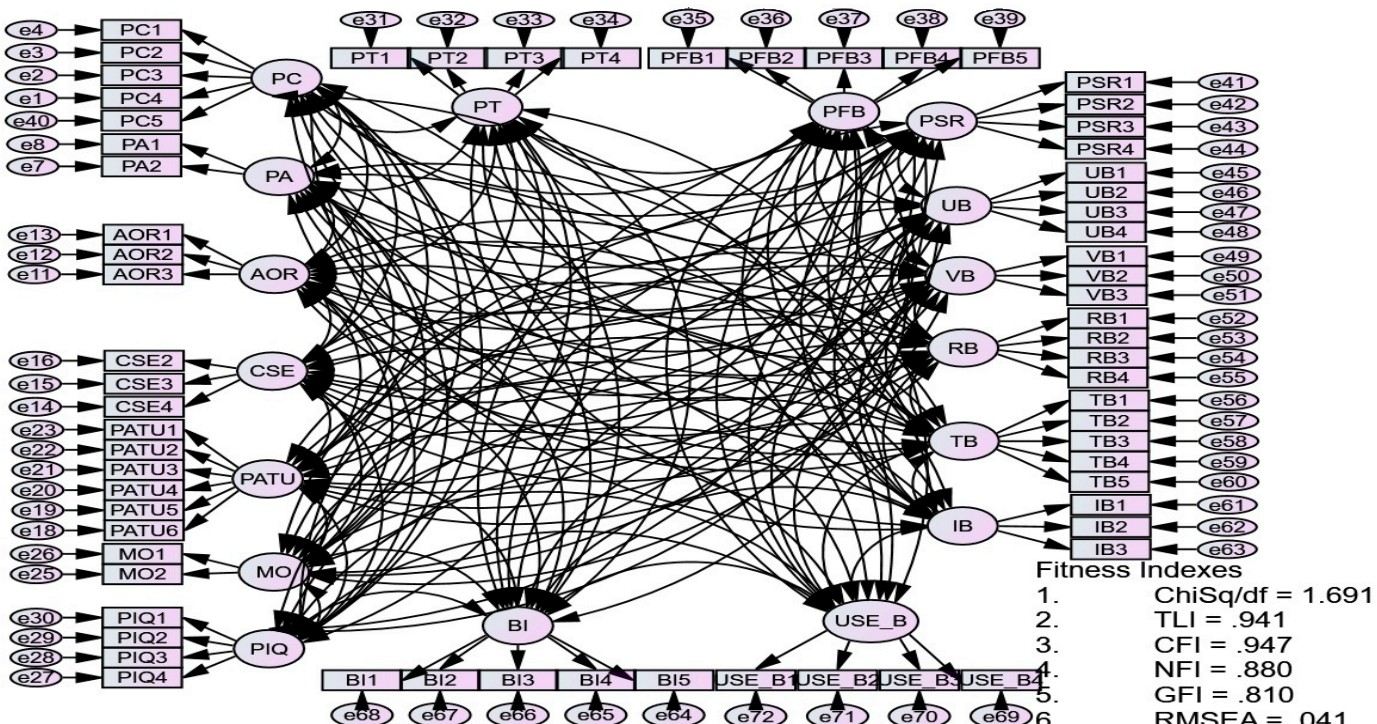

**Figure 4.** Measurement model.

**Table 2.** Summary of measurement model reports.

| Construct | Items | Factor Loading | Cronbach's Alpha (≥0.7) | CR (≥0.7) | AVE (≥0.5) |
|---|---|---|---|---|---|
| PC | PC1 | 0.664 | 0.875 | 0.874 | 0.584 |
| | PC2 | 0.725 | | | |
| | PC3 | 0.712 | | | |
| | PC4 | 0.868 | | | |
| | PC5 | 0.834 | | | |
| PA | PA1 | 0.813 | 0.812 | 0.827 | 0.705 |
| | PA2 | 0.865 | | | |
| AOR | AOR1 | 0.791 | 0.808 | 0.806 | 0.501 |
| | AOR2 | 0.677 | | | |
| | AOR3 | 0.648 | | | |
| CSE | CSE2 | 0.717 | 0.891 | 0.894 | 0.740 |
| | CSE3 | 0.908 | | | |
| | CSE4 | 0.939 | | | |
| PATU | PATU1 | 0.849 | 0.933 | 0.932 | 0.697 |
| | PATU2 | 0.859 | | | |
| | PATU3 | 0.842 | | | |
| | PATU4 | 0.836 | | | |
| | PATU5 | 0.862 | | | |
| | PATU6 | 0.755 | | | |
| MO | MO1 | 0.859 | 0.862 | 0.871 | 0.772 |
| | MO2 | 0.898 | | | |
| PIQ | PIQ1 | 0.835 | 0.931 | 0.932 | 0.775 |
| | PIQ2 | 0.909 | | | |
| | PIQ3 | 0.903 | | | |
| | PIQ4 | 0.872 | | | |
| PT | PT1 | 0.838 | 0.896 | 0.902 | 0.699 |
| | PT2 | 0.730 | | | |
| | PT3 | 0.873 | | | |
| | PT4 | 0.894 | | | |

**Table 2.** *Cont.*

| Construct | Items | Factor Loading | Cronbach's Alpha (≥0.7) | CR (≥0.7) | AVE (≥0.5) |
|---|---|---|---|---|---|
| PFB | PFB1 | 0.873 | | | |
| | PFB2 | 0.902 | | | |
| | PFB3 | 0.920 | 0.934 | 0.956 | 0.812 |
| | PFB4 | 0.920 | | | |
| | PFB5 | 0.891 | | | |
| PSR | PSR1 | 0.831 | | | |
| | PSR2 | 0.868 | 0.896 | 0.881 | 0.651 |
| | PSR3 | 0.747 | | | |
| | PSR4 | 0.775 | | | |
| UB | UB1 | 0.601 | | | |
| | UB2 | 0.903 | 0.899 | 0.902 | 0.709 |
| | UB3 | 0.893 | | | |
| | UB4 | 0.928 | | | |
| VB | VB1 | 0.862 | | | |
| | VB2 | 0.903 | 0.909 | 0.908 | 0.767 |
| | VB3 | 0.861 | | | |
| RB | RB1 | 0.817 | | | |
| | RB2 | 0.902 | 0.914 | 0.914 | 0.727 |
| | RB3 | 0.888 | | | |
| | RB4 | 0.800 | | | |
| TB | TB1 | 0.833 | | | |
| | TB2 | 0.853 | | | |
| | TB3 | 0.902 | 0.939 | 0.936 | 0.744 |
| | TB4 | 0.876 | | | |
| | TB5 | 0.847 | | | |
| IB | IB1 | 0.918 | | | |
| | IB2 | 0.935 | 0.918 | 0.922 | 0.798 |
| | IB3 | 0.823 | | | |
| BI | BI1 | 0.765 | | | |
| | BI2 | 0.795 | | | |
| | BI3 | 0.854 | 0.912 | 0.910 | 0.669 |
| | BI4 | 0.845 | | | |
| | BI5 | 0.826 | | | |
| USE-B | USE-B1 | 0.792 | | | |
| | USE-B2 | 0.726 | 0.864 | 0.865 | 0.615 |
| | USE-B3 | 0.834 | | | |
| | USE-B4 | 0.782 | | | |

Notes: PC = Perceived Compatibility, PA = Perceived Awareness, AOR = Availability of Resources, CSE = Computer Self Efficacy, PATU = Perceived Ability to Use, MO = Multilingual Option, PIQ = Perceived Information Quality, PT = Perceived Trust, PFB = Perceived Functional Benefit, PSR = Perceived Service Response, UB = Usage Barrier, VB = Value Barrier, RB = Risk Barrier, TB = Tradition Barrier, IB = Image Barrier, BI = Behavioral Intention, USE-B = Use Behavior.

**Table 3.** Discriminate validity.

| | PC | PA | AOR | CSE | PATU | MO | PIQ | PT | PFB | PSR | UB | VB | RB | TB | IB | BI | USE-B |
|---|---|---|---|---|---|---|---|---|---|---|---|---|---|---|---|---|---|
| **PC** | **0.765** | | | | | | | | | | | | | | | | |
| **PA** | 0.759 | **0.839** | | | | | | | | | | | | | | | |
| **AOR** | 0.642 | 0.639 | **0.708** | | | | | | | | | | | | | | |
| **CSE** | 0.648 | 0.645 | 0.625 | **0.860** | | | | | | | | | | | | | |
| **PATU** | 0.704 | 0.712 | 0.703 | 0.821 | **0.835** | | | | | | | | | | | | |
| **MO** | 0.543 | 0.590 | 0.606 | 0.530 | 0.628 | **0.878** | | | | | | | | | | | |
| **PIQ** | 0.717 | 0.675 | 0.705 | 0.630 | 0.820 | 0.654 | **0.880** | | | | | | | | | | |
| **PT** | 0.694 | 0.641 | 0.607 | 0.622 | 0.750 | 0.666 | 0.803 | **0.836** | | | | | | | | | |
| **PFB** | 0.593 | 0.482 | 0.468 | 0.447 | 0.547 | 0.504 | 0.558 | 0.600 | **0.901** | | | | | | | | |
| **PSR** | 0.549 | 0.494 | 0.616 | 0.486 | 0.671 | 0.402 | 0.696 | 0.583 | 0.413 | **0.807** | | | | | | | |
| **UB** | 0.145 | 0.130 | 0.078 | 0.134 | 0.181 | 0.205 | 0.142 | 0.203 | 0.235 | 0.018 | **0.842** | | | | | | |
| **VB** | 0.140 | 0.124 | 0.076 | 0.146 | 0.186 | 0.131 | 0.102 | 0.153 | 0.171 | 0.038 | 0.704 | **0.875** | | | | | |
| **RB** | 0.173 | 0.186 | 0.105 | 0.160 | 0.172 | 0.107 | 0.102 | 0.196 | 0.174 | 0.037 | 0.757 | 0.760 | **0.853** | | | | |

**Table 3.** *Cont.*

|  | PC | PA | AOR | CSE | PATU | MO | PIQ | PT | PFB | PSR | UB | VB | RB | TB | IB | BI | USE-B |
|---|---|---|---|---|---|---|---|---|---|---|---|---|---|---|---|---|---|
| **TB** | 0.217 | 0.162 | 0.154 | 0.184 | 0.260 | 0.130 | 0.194 | 0.203 | 0.199 | 0.149 | 0.815 | 0.709 | 0.775 | **0.863** | | | |
| **IB** | 0.136 | 0.200 | 0.054 | 0.168 | 0.175 | 0.115 | 0.103 | 0.192 | 0.172 | 0.048 | 0.748 | 0.727 | 0.747 | 0.786 | **0.893** | | |
| **BI** | 0.739 | 0.621 | 0.703 | 0.640 | 0.768 | 0.643 | 0.741 | 0.817 | 0.642 | 0.657 | 0.230 | 0.190 | 0.209 | 0.235 | 0.808 | **0.818** | |
| **USE-B** | 0.675 | 0.549 | 0.598 | 0.577 | 0.686 | 0.498 | 0.683 | 0.633 | 0.747 | 0.555 | 0.192 | 0.165 | 0.179 | 0.227 | 0.694 | 0.782 | **0.784** |

Notes: PC = Perceived Compatibility, PA = Perceived Awareness, AOR = Availability of Resources, CSE = Computer Self Efficacy, PATU = Perceived Ability to Use, MO = Multilingual Option, PIQ = Perceived Information Quality, PT = Perceived Trust, PFB = Perceived Functional Benefit, PSR = Perceived Service Response, UB = Usage Barrier, VB = Value Barrier, RB = Risk Barrier, TB = Tradition Barrier, IB = Image Barrier, BI = Behavioral Intention, USE-B = Use Behavior.

### 4.4. Structural Model

Figure 5 shows that the Goodness of Fit (GOF) indices provide a solid foundation for the proposed model's excellent data fit. Both the GFI and the NFI suggest a reading above 0.80, whereas the TLI and the CFI suggest a reading above 0.90. Additionally, the ChiSq/df is 3, and the RMSEA is less than 0.08. Table 4 displays the outcomes of the hypothesis test.

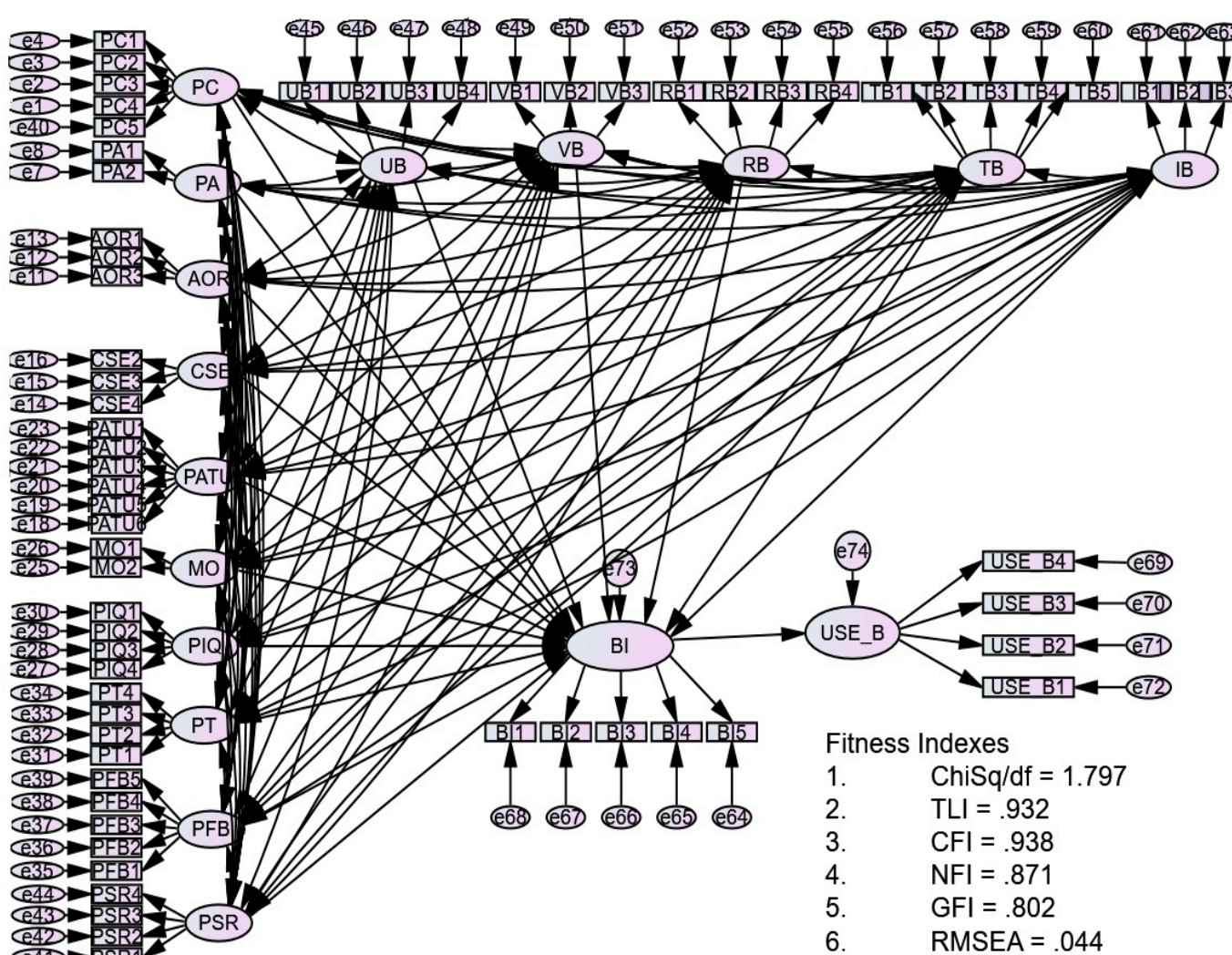

**Figure 5.** Structural model.

**Table 4.** Results of hypothesis testing.

|  | Hypothesis | *p* | Result |
|---|---|---|---|
| H1 | PC→BI | 0.003 | Supported |
| H2 | PA→BI | 0.031 | Supported |
| H3 | AOR→BI | *** | Supported |
| H4 | CSE→BI | 0.412 | Not supported |
| H5 | PATU→BI | 0.687 | Not supported |
| H6 | MO→BI | 0.515 | Not supported |
| H7 | PIQ→BI | 0.013 | Supported |
| H8 | PT→BI | *** | Supported |
| H9 | PFB→BI | *** | Supported |
| H10 | PSR→BI | *** | Supported |
| H11 | UB→BI | 0.471 | Not supported |
| H12 | VB→BI | 0.501 | Not supported |
| H13 | RB→BI | 0.790 | Not supported |
| H14 | TB→BI | 0.105 | Not supported |
| H15 | IB→BI | 0.332 | Not supported |
| H16 | BI→USE-B | *** | Supported |

Notes: PC = Perceived Compatibility, PA = Perceived Awareness, AOR = Availability of Resources, CSE = Computer Self Efficacy, PATU = Perceived Ability to Use, MO = Multilingual Option, PIQ = Perceived Information Quality, PT = Perceived Trust, PFB = Perceived Functional Benefit, PSR = Perceived Service Response, UB = Usage Barrier, VB = Value Barrier, RB = Risk Barrier, TB = Tradition Barrier, IB = Image Barrier, BI = Behavioral Intention, USE-B = Use Behavior, *** = smaller than 0.001.

## 5. Discussion

The results of this study demonstrated that the behavioral intention to use digital public services is directly and favorably impacted by perceived compatibility. This result supported hypothesis H1 and is consistent with earlier research (e.g., Dubey and Sahu 2022; Kumar and Sachan 2017; Mandari and Chong 2018; Moqbel et al. 2014; Shetu et al. 2022; Todeschini et al. 2020; Tung et al. 2014; Wang et al. 2022). This indicates that most of those who use electronic public services in Saudi Arabia can find the information they need from the electronic public services sites and that the public electronic services sites are suitable for their needs. They would rather communicate digitally with automated public service websites than in person with office personnel. Their use of public service websites is consistent with the way they like to interact. In general, the use of electronic public service websites is commensurate with their lifestyle.

The results also demonstrated that the behavioral intention to use digital public services is directly and favorably impacted by perceived awareness, which supported Hypothesis H2 and is consistent with earlier research (e.g., Abdurakhimovna et al. 2021; Alzoubi et al. 2019; Hasan et al. 2018; Mason and Nassivera 2013; Pantari and Aji 2020; Yuan and Jang 2008). This means that most users of electronic public service websites in Saudi Arabia are familiar with these websites and know the benefits of using them. There are government campaigns in Saudi Arabia that inform citizens about the advantages of using e-government services online, and there are also training programs that teach citizens how to utilize these sites effectively. This study's findings indicate that the availability of resources has an immediate and favorable effect on the behavioral intention to use digital public services.

Hypothesis H3 also found support in the results, which is consistent with earlier research (e.g., Abdel-Wahab 2008; Dubey and Sahu 2022; Eke 2011; Hasan et al. 2018). Those who rely on electronic services offered by the government can access those services from any location, including their homes and places of employment, provided they have access to the internet. Further, people can obtain reasonably priced high-speed internet access both at their homes and at their places of employment, and they take pleasure in the continual availability of a high-speed internet connection. The results demonstrated that the behavioral intention to use digital public services is directly and favorably impacted by perceived information quality.

Hypothesis H7 was also supported by the results, which is in line with earlier research (e.g., Aryani et al. 2022; Tung et al. 2014; Zhang and Xi 2022; Zardari et al. 2021). Thus, people who utilize the services provided by the government online believe that the information is up-to-date and easy to understand. These sites provide all the necessary relevant information and accurate information about the services provided by the site. Further, we found that the behavioral intention to use digital public services is directly and favorably affected by perceived trust. This result supports Hypothesis H8, which is consistent with earlier research (e.g., Alzoubi et al. 2019; Bélanger and Carter 2008; Chatzoglou et al. 2015; Gu et al. 2009; Koundinya 2019; Kumar and Sachan 2017; Voutinioti 2013; Zahid and Din 2019; Zeebaree et al. 2022).

This implies that those who utilize websites to obtain public services have more faith in these locations than they do in their local government offices, and they think the websites are more secure than the offices themselves. Most users think it is possible to trust official websites that offer online services.

The results of this study further demonstrated that the behavioral intention to use digital public services is directly and favorably impacted by perceived functional benefits. This result supports Hypothesis H9, as previously indicated (e.g., Alghamdi and Beloff 2014; Alghamdi and Beloff 2016). Users of digital public services believe that the most significant advantages of these sites are that they can be accessed from anywhere and at any time and that using e-government services websites is cheaper than visiting the actual government office. Submitting a service request using one of the many websites currently accessible to deliver e-government services is much quicker than the traditional method of receiving government services.

We also showed that behavioral intention to use digital public services is directly and favorably impacted by perceived service response, which is in support of Hypothesis H10 and in line with earlier research (e.g., Ahmad and Sahari 2011; Kumar and Sachan 2017; Li and Shang 2020). Users of digital public service websites believe that customer service on e-government service websites takes immediate assistance measures when encountering any problem and that online customer service is available at all times and responds very quickly, which increases the intent to use these websites. In support of Hypothesis H16, the use behavior of digital public services was found to be directly and favorably affected by behavioral intention, as demonstrated in earlier research (e.g., ElKheshin and Saleeb 2020; Khurshid et al. 2022; Rana et al. 2012; Sahari et al. 2012; Shyu and Huang 2011).

This means that behavioral intent is an important factor influencing usage behavior. It was found through the respondents' answers that they regularly use the services available on government websites and that they search for information about government services through their web pages. They can also easily access e-government services. In general, they prefer to use the services available on government websites.

Findings from this study also show that people's propensity to utilize digital public services is not affected by characteristics such as their level of computer self-efficacy, their confidence in their ability to use the services effectively, or the availability of services in several languages. This indicates that Hypotheses H4, H5, and H6 were rejected.

Moreover, certain barriers are mentioned in IRT, although those barriers did not affect the intention to use digital public services in this study. We found that the respondents' intent to use digital public services was unaffected by usage barriers, value barriers, risk barriers, tradition barriers, or image barriers. This indicates the rejection of Hypotheses H11, H12, H13, H14, and H15.

A number of obstacles are noted using the framework of innovation resistance theory (IRT) that may prevent people from embracing new technologies, like digital public services. However, it was shown that in this study, these constraints had no appreciable influence on consumers' intentions to use digital public services. The empirical results of this study did not support IRT's hypothesis that usage barriers, value barriers, risk barriers, tradition barriers, and image barriers might prevent adoption. This shows that the variables that

could ordinarily dissuade people from using digital public services were not significant enough to discourage users in the studied scenario.

## 6. Implications

### 6.1. Theoretical Implications

The majority of past research on the subject of digital public services with respect to Saudi Arabia has not focused on integrating variables from the GAM and IRT. Even so, the goal of this study was to combine the GAM and IRT parts into a single model. This research examines the effects of well-known models (GAM and IRT) and structures on the accessibility of digital public services. This study could potentially serve as reference research to inform future researchers in the field of digital public services due to the lack of studies completed in the setting of Saudi Arabia. This study's findings contribute to the growing body of empirical work on the topic. The results of this study suggest that several elements, each with their own influence, contribute to the establishment of behavioral intentions to utilize digital public services. Some of these include perceived compatibility, awareness, availability of resources, quality of information, trust, functional benefits, and service response. The importance of the current study lies in its proposal to merge the GAM and IRT and in its linking of these two theories to the concept of digital public services.

Drawing on an integrated model helps enhance understanding of the ways in which such variables might contribute to digital (ex)inclusion. This is of course not to reduce a complex phenomenon such as digital inclusion to a limited set of variables; this would be falling into the traps of technologically deterministic thinking. This study acknowledges that using such models while limiting what we can see and study also channels focus to important aspects of this phenomenon that otherwise might be lost if we try to study everything at once. We can build accumulative knowledge only through producing multiple studies that each focus on one important aspect. This study comes in the spirit of an accumulative tradition, and it does this by integrating the GAM and IRT in a novel way.

### 6.2. Practical Implications

Using a research framework built on a solid theoretical and literature review foundation, this study elucidates the underlying concepts that can inform digital inclusion policy design. We conducted an empirical study in Saudi Arabia and used rigorous statistical analysis to verify our GAM and IRT models across six categories of inclusion in digital public services in Saudi Arabia. From these results, it is obvious that citizens in Saudi Arabia play a pivotal role in determining which factors are most important for the widespread inclusion of digital public services and in pinpointing the consequences of implementing varying degrees of digital public service inclusion.

Digital public service inclusion in Saudi Arabia will fail to replace the old brick-and-mortar government system unless the needs and fundamental desires of citizens to accept digital public services are met. This study confirmed two distinct inclusion models—the GAM and IRT—that may be thoroughly researched and employed in the creation of a digital public services framework with the citizen at its center. The key source of success for both government agencies and citizens is the adoption of a framework that includes digital public services at different levels in Saudi Arabia.

Therefore, policymakers in Saudi Arabia should understand that attitudes toward use, ability to use, assurance of use, adherence to use, adaptability, functional barriers, and psychological barriers are potential contributors to digital public service inclusion when implementing digital public services and setting strategic initiatives to develop digital public services.

## 7. Conclusions

The goal of this study was to identify the factors that can make government-provided digital services more inclusive by developing a model based on a synthesis of two theories (GAM and IRT). The quantitative approach we took involved using the results of

previous studies to guide the creation of a questionnaire with 72 questions, the validity and reliability of which were determined to be high. The findings indicate that perceived compatibility, awareness, availability of resources, perceived information quality, perceived trust, perceived functional benefits, and perceived service responsiveness are the most important elements contributing to digital inclusion in the context of e-government. The study's model can be validated in other countries or contexts to support the generalization of the study's conclusions.

One of the limitations of this study is that it focused on the factors that lead to the inclusion of digital public services from the perspective of citizens. Another limitation of this study is that it focused on citizens in Saudi Arabia; it was a single-country case study. Moreover, this study relied heavily on a quantitative approach using variance models instead of a qualitative process-oriented approach. Future studies can focus on studying digital inclusion using, for example, a process theory or a phenomenological approach, ideally in multiple contexts, to enhance our overall understanding of how digital inclusion is enacted in practice.

**Author Contributions:** Conceptualization, Y.H.A.-M. and M.A.; methodology, Y.H.A.-M.; software, Y.H.A.-M.; validation, Y.H.A.-M. and M.A.; formal analysis, Y.H.A.-M.; investigation, Y.H.A.-M. and M.A.; resources, Y.H.A.-M. and M.A.; data curation, Y.H.A.-M. and M.A.; writing—original draft preparation, Y.H.A.-M. and M.A.; writing—review and editing, Y.H.A.-M. and M.A.; visualization, Y.H.A.-M. and M.A.; supervision, Y.H.A.-M.; project administration, Y.H.A.-M.; funding acquisition, Y.H.A.-M. and M.A. All authors have read and agreed to the published version of the manuscript.

**Funding:** No funding was received for conducting this study.

**Institutional Review Board Statement:** The study was conducted in accordance with the Declaration of Helsinki, and approved by the Research Ethics Committee (REC) at University of Hail dated: 2/10/2023 code H-2023-363.

**Informed Consent Statement:** Informed consent was obtained from all subjects involved in the study.

**Data Availability Statement:** Data will be available upon request.

**Conflicts of Interest:** The authors declare no conflict of interest.

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
