# Peer review of "Making Digital Government More Inclusive: An Integrated Perspective"

_socsci, doi:10.3390/socsci12100557_

Round 1
Reviewer 1 Report
This manuscript delves into e-government in Saudi Arabia, which is a timely topic of discussion. Upon reviewing the abstract, it becomes clear that the manuscript is well-written, effectively highlighting the research problem, aim, methodology, and primary findings. I have some comments that may help improve this study.
1- The introduction needs to be restructured to explain the gap in the literature with supportive references, the research problem, research questions, aim/purpose, methods in brief, and contribution. I recommend having a paragraph showing each of the items mentioned here.
2- There are too many titles for these manuscripts; the reader must read five titles to understand the research methods. I suggest summarizing titles starting from 2 till 4 and adding them under the main title, "literature review."
3- More clarification is required for the research method regarding the survey questions and the parameters for the sample size.
4- The discussion section should be extended. It would be beneficial to connect the current outcomes with prior research. I would recommend an extended explanation in such linkages with specific what is mentioned in Lines 572–579 on Page 18. On page 18, Lines 600-604, the author/s discusses the challenges of IRT. Additionally, it is essential to discuss the limitations of this study and the method/s used.
5- The conclusion section should have a paragraph or two suggesting future research based on the research limitations.
The English is good.
Author Response
|
1- The introduction needs to be restructured to explain the gap in the literature with supportive references, the research problem, research questions, aim/purpose, methods in brief, and contribution. I recommend having a paragraph showing each of the items mentioned here. Answer to comment Added please check page 1 and page 2 (Highlighted in red color) |
|
2- There are too many titles for these manuscripts; the reader must read five titles to understand the research methods. I suggest summarizing titles starting from 2 till 4 and adding them under the main title, "literature review." Answer to comment Done |
|
3- More clarification is required for the research method regarding the survey questions and the parameters for the sample size. Answer to comment Added please check 3.2. Research Instrument and Measures 3.1. Research Design |
|
4- The discussion section should be extended. It would be beneficial to connect the current outcomes with prior research. I would recommend an extended explanation in such linkages with specific what is mentioned in Lines 572–579 on Page 18. On page 18, Lines 600-604, the author/s discusses the challenges of IRT. Additionally, it is essential to discuss the limitations of this study and the method/s used. Answer to comment The authors connect the current outcomes with prior research please check the discussion section from 543 to line 611. The authors discusses the challenges of IRT please check line 621 to line 628. |
|
5- The conclusion section should have a paragraph or two suggesting future research based on the research limitations. Answer to comment Added |
Reviewer 2 Report
Thank you for the opportunity to read this interesting paper. I enjoyed reading it, but found some major flaws which need to be addressed by the authors in order from them to strengthen their arguments.
On p2 you introduce the notion of " digitally disadvantaged individuals" and that these need to be supported in order able to fully reap the benefits of the internet. You then inlcude your understanding of digital inclusion, in particular in the context of e-government, but:
This choice of words raises the following questions:
1. what is a digitally disadvantaged individual? Can we compare this to notions such as physically/sociall/financially disadvantaged individulas?
2. What is your definition for digitally disadvantaged individuals? I detect a technological assumption in your way of presenting the case - in other words, digital inclusion is a always beneficial?
The GAM is very technologically deterministic, what are the advantages of this model, what are the disadvantages? Would it not be an advantage to add some other key variables that may lead to digital exclusion/disadvantage?
E-government is no longer a "new" discipline, several scholars such as Scholl, Yildiz, Bannister have shown that is an interdisciplinary and transdisciplinary discipline. It is not at a"conceptual" or "theoretical" - on the contrary, it stems from the attempts and problems associated with bringing technology into government and public administration. Rather, the difficulty is to link practice and theory.
Not all agree that e-government is just about imporiving service quality. Whilst it is certainly one aim, there are many more, including PA reform, organisational change, reducing costs, increasing transparency, efficiency etc. Also, it may not only be citizen-focused as citizens may be one stakeholder, but there are many more, oftentimes political (see e.g. work by Soares on the role of e-gov benchmarks).
I do agree that more empirical is necessary, but I think your background on e-government "history" needs to be revised, more critical and include more by key scholars (as mentioned above, or see also Janssen, Parycek, Wimmer. Pardo, Gil-Garcia, Janowski, Estevez, Pereira-Viale).
Whilst I find the use of the Innovation Resistance Theory very interesting, this is juxtaposing your previous technologically deterministic approach to digital inclusion with human barriers to adoption. I would recommend reading "Bannister, F., & Connolly, R. (2020). The future ain't what it used to be: Forecasting the impact of ICT on the public sphere. Government Information Quarterly, 37(1)" in order to gain an alternative understand why technology/e-government forecasts fail.
Data collection.
This study aims to collect data in order to find out how to provide digital services to all. The data is collected online using a google form. Does this not exclude exactly the kind of individuals you are trying to collect data from (those who are digitally included)? If the aim of e-government is to be citizen-centric, then data must include those citizens who are unable to access public digital services.
In the light of this major methodological flaw (unless it can be well-explained by the author/s), I am not sure how to understand/appreicate the results gained. Unless the aim is to support the use of the GAM model (p.19) which is not the aim stated at the end of the literature review.
I would also encourage to present the Kongdom of Saudi Arabia as a particular case ("casing" in the methodological section or as a "case study") so that it is possible to highlight the advantages and limitations in the conclusion section - in particular to other countries, as well as the lessons learned for other countries and e-government stakeholders.
Author Response
|
On p2 you introduce the notion of " digitally disadvantaged individuals" and that these need to be supported in order able to fully reap the benefits of the internet. You then inlcude your understanding of digital inclusion, in particular in the context of e-government, but:
This choice of words raises the following questions:
1. what is a digitally disadvantaged individual? Can we compare this to notions such as physically/sociall/financially disadvantaged individulas?
2. What is your definition for digitally disadvantaged individuals? I detect a technological assumption in your way of presenting the case - in other words, digital inclusion is a always beneficial? Answer to comment Thank you for this important note. It is true that the word should be deleted because it raises many questions and replaced with the word individuals |
|
The GAM is very technologically deterministic, what are the advantages of this model, what are the disadvantages? Would it not be an advantage to add some other key variables that may lead to digital exclusion/disadvantage? Answer to comment
Thank you for your valuable comment. Advantages of the GAM's technological determinism include its focus on technology's direct influence, aiding clear understanding of adoption. |
|
E-government is no longer a "new" discipline, several scholars such as Scholl, Yildiz, Bannister have shown that is an interdisciplinary and transdisciplinary discipline. It is not at a"conceptual" or "theoretical" - on the contrary, it stems from the attempts and problems associated with bringing technology into government and public administration. Rather, the difficulty is to link practice and theory. Answer to comment Thank you for your valuable comment. In the context of the Kingdom of Saudi Arabia, the concept of e-government and the automation of public services is considered an important development and step in the field of technology |
|
Not all agree that e-government is just about imporiving service quality. Whilst it is certainly one aim, there are many more, including PA reform, organisational change, reducing costs, increasing transparency, efficiency etc. Also, it may not only be citizen-focused as citizens may be one stakeholder, but there are many more, oftentimes political (see e.g. work by Soares on the role of e-gov benchmarks). Answer to comment Thank you for your valuable comment. Added please check section 7. Conclusions
|
|
Whilst I find the use of the Innovation Resistance Theory very interesting, this is juxtaposing your previous technologically deterministic approach to digital inclusion with human barriers to adoption. I would recommend reading "Bannister, F., & Connolly, R. (2020). The future ain't what it used to be: Forecasting the impact of ICT on the public sphere. Government Information Quarterly, 37(1)" in order to gain an alternative understand why technology/e-government forecasts fail. Answer to comment
The Innovation Resistance Theory (IRT) is particularly intriguing in the area of e-government since it can reveal covert obstacles and difficulties that prevent the adoption of digital advances. IRT investigates the complex reasons why people might be reluctant to adopt new technologies, providing light on elements that go beyond merely accessibility or usability. Understanding the factors that contribute to citizens' reluctance or resistance to adopt digital services can help to inform the design of more efficient, user-centered, and inclusive e-government initiatives in the context of e-government, where technological advancements frequently have policy and societal implications. IRT's findings help to clarify the intricacies of digital inclusion and lay the groundwork for specific approaches to deal with the issues that can obstruct the successful adoption of e-government services. |
|
Data collection.
This study aims to collect data in order to find out how to provide digital services to all. The data is collected online using a google form. Does this not exclude exactly the kind of individuals you are trying to collect data from (those who are digitally included)? If the aim of e-government is to be citizen-centric, then data must include those citizens who are unable to access public digital services.
In the light of this major methodological flaw (unless it can be well-explained by the author/s), I am not sure how to understand/appreicate the results gained. Unless the aim is to support the use of the GAM model (p.19) which is not the aim stated at the end of the literature review.
I would also encourage to present the Kongdom of Saudi Arabia as a particular case ("casing" in the methodological section or as a "case study") so that it is possible to highlight the advantages and limitations in the conclusion section - in particular to other countries, as well as the lessons learned for other countries and e-government stakeholders. Answer to comment
Added please check 3.3. Data Collection and Analysis Procedures Also limitations added in the conclusion section
|
Round 2
Reviewer 1 Report
Thank your for addressing all my comments and provide a revised version.
Author Response
Thank you very much for accepting my paper for publication. Your consideration is greatly appreciated.
Reviewer 2 Report
Thank you for trying to address some of the issues I highlighted.
Although there is a clear attempt at addressing them, I still think that there are still some important aspects that the authors need to consider.
1. Technological/digital determinism
This continues to be a very technologically deterministic paper, and although I urged the authors to address the disadvantages of such an approach/the GAM I do not see the authors discussing the disadvantages of technological determinism. I again encourage the authors to take a more balanced/cirital view, e.g. by considering the article mentioned by Bannister in my first review.
2. Even if e-government is "new" in Saudia Arabia, it needs to consider the previous scholarship that has lead to e-government in its current form and theory worldwide.
3. I am still missing the casing of Saudi Arabia in the methods section (see work by Yin on how to present it).
Author Response
Response to reviewer 2:
Thank you for your developmental comments. We have addressed the issue regarding technological determinism. We do not subscribe to a technologically deterministic worldview neither that we use GAM and IRT in this spirit. Given this, we also cannot ignore the materiality of technology and how material features of an artefact do, to some extent, constrain the agency of actors as they interact with it. We have added several paragraphs in the theoretical framework section as well as in the theoretical implications delineating how we use GAM and IRT not in a technologically deterministic way, rather building on an accumulative tradition where we build on previous studies and theories to identify the conditions that might contribute to digital inclusion. The scope of this study does not include studying the social or cultural conditions that might underpin digital inclusion. We believe in an eclectic approach that employees several methodologies to reach a more nuanced understanding. In this study, we adopt a quantitative approach whereby we integrate two of the most prominent theories in the field in a novel way. We have another qualitative study that builds on an interpretative approach that is currently under review at another journal where study the same phenomenon using different tools and approaches. Below, we quote some of the newly added texts in the manuscript to make it easy for you to refer to it:
“The E-Government Adoption Model (GAM) can be criticized as underpinned by an unescapable technological deterministic underpinning in that it narrows down the complex multifaceted phenomena of e-government adoption to specific measurable variables (Grint & Woolgar, 1997). This, nevertheless, does not necessarily mean that such variables are irrelevant (Sayer, 1997). Also, the materiality of technology cannot be ignored (Leonardi & Barely, 2008). In the real world, the materiality of everyday life does constrain human agency in varied ways; think of how the designed features of a technology could constrain how people interact with that technological artefact (Leonardi, 2011). Moreover, as Winner (1980) had shown, the designers of technology do, indeed, imbed their world views in how they build their artefacts thereby using the materiality of technology to further their objectives. Given that technological artefacts have design features and that such features could constrain human agency, it becomes imperative to study how such material features could, in this case, impede or foster digital inclusion. In this study, we adopt what might be thought of as technologically deterministic models, not to further a technologically deterministic view, but to study and be aware of how the materiality of technology could play a role in digital inclusion. Models such as GAM or Innovation Resistance Theory (IRT) can be valuable tools when adopted with care and attention to the pitfalls of technological determinism. In what follows, we explore how GAM has been used in the e-government literature and how it might help inform the digital inclusion debate.”
With respect to Bannister and Connolly’s (2020) arguments regarding why forecasts fail. We strongly agree with their perspective. Reading the article, it does not come across as they are advocating that researchers stop forecasting, especially regarding the future impacts of technology. They do, nevertheless, provide guardrails and guidelines on how to not fall into the pitfalls of forecasting they identified. In this study, our aim is not to forecast but to understand what factors might contribute to digital inclusion.